# Transforming RNA-Seq gene expression to track cancer progression in the multi-stage early to advanced-stage cancer development

Michelle Livesey[1], Sophia Catherine Rossouw[1], Renette Blignaut[2], Alan Christoffels[1], Hocine Bendou[1]*

1 South African Medical Research Council Bioinformatics Unit, South African National Bioinformatics Institute, University of the Western Cape, Cape Town, South Africa, 2 Department of Statistics and Population Studies, University of the Western Cape, Cape Town, South Africa

* hocine@sanbi.ac.za

## Abstract

### Background

Cancer progression can be tracked by gene expression changes that occur throughout early-stage to advanced-stage cancer development. The accumulated genetic changes can be detected when gene expression levels in advanced-stage are less variable but show high variability in early-stage. Normalizing advanced-stage expression samples with early-stage and clustering of the normalized expression samples can reveal cancers with similar or different progression and provide insight into clinical and phenotypic patterns of patient samples within the same cancer.

### Objective

This study aims to investigate cancer progression through RNA-Seq expression profiles across the multi-stage process of cancer development.

### Methods

RNA-sequenced gene expression of Diffuse Large B-cell Lymphoma, Lung cancer, Liver cancer, Cervical cancer, and Testicular cancer were downloaded from the UCSC Xena database. Advanced-stage samples were normalized with early-stage samples to consider heterogeneity differences in the multi-stage cancer progression. WGCNA was used to build a gene network and categorized normalized genes into different modules. A gene set enrichment analysis selected key gene modules related to cancer. The diagnostic capacity of the modules was evaluated after hierarchical clustering.

### Results

Unnormalized RNA-Seq gene expression failed to segregate advanced-stage samples based on selected cancer cohorts. Normalization with early-stage revealed the true hetero-geneous gene expression that accumulates across the multi-stage cancer progression, this resulted in well segregated cancer samples. Cancer-specific pathways were enriched in the

**Data Availability Statement:** The RNA-Seq gene expression and curated clinical public datasets underlying the results presented in the study are

available from the following direct URLs: TCGA-DLBC (https://gdc-hub.s3.us-east-1.amazonaws.com/download/TCGA-DLBC.htseq_counts.tsv.gz and https://toil-xena-hub.s3.us-east-1.amazonaws.com/download/GTEX_phenotype.gz),TCGA-LUAD (https://gdc-hub.s3.us-east-1.amazonaws.com/download/TCGA-LUAD.htseq_counts.tsv.gz and https://gdc-hub.s3.us-east-1.amazonaws.com/download/TCGA-LUAD.GDC_phenotype.tsv.gz), TCGA-LIHC (https://gdc-hub.s3.us-east-1.amazonaws.com/download/TCGA-LIHC.htseq_counts.tsv.gz and https://gdc-hub.s3.us-east-1.amazonaws.com/download/TCGA-LIHC.GDC_phenotype.tsv.gz), TCGA-CESC (https://gdc-hub.s3.us-east-1.amazonaws.com/download/TCGA-CESC.htseq_counts.tsv.gz and https://gdc-hub.s3.us-east-1.amazonaws.com/download/TCGA-CESC.GDC_phenotype.tsv.gz), and TCGA-TGCT (https://gdc-hub.s3.us-east-1.amazonaws.com/download/TCGA-TGCT.htseq_counts.tsv.gz and https://gdc-hub.s3.us-east-1.amazonaws.com/download/TCGA-TGCT.GDC_phenotype.tsv.gz). The normal tissue were obtained from the Genotype-Tissue Expression (GTEx) Portal as follows: https://toil-xena-hub.s3.us-east-1.amazonaws.com/download/gtex_gene_expected_count.gz and https://toil-xena-hub.s3.us-east-1.amazonaws.com/download/GTEX_phenotype.gz, for the expression and phenotype information, respectively (S1 File).

**Funding:** This work was supported by the South African Medical Research Council and National Research Foundation of South Africa grant: 121787 (M.L). https://www.samrc.ac.za & http://www.nrf.ac.za. The funders had no role in study design, data collection and analysis, decision to publish, or preparation of the manuscript.

**Competing interests:** The authors have declared that no competing interests exist.

normalized WGCNA modules. The normalization method was further able to stratify patient samples based on phenotypic and clinical information. Additionally, the method allowed for patient survival analysis, with the Cox regression model selecting gene MAP4K1 in cervical cancer and Kaplan-Meier confirming that upregulation is favourable.

## Conclusion

The application of the normalization method further enhanced the accuracy of clustering of cancer samples based on how they progressed. Additionally, genes responsible for cancer progression were discovered.

## 1. Introduction

Cancer is an ever-changing disease that generally becomes more heterogeneous as the disease progresses [1]. Different cancers progress and evolve in different ways. Some cancers are fast-growing and can cause mortality soon after diagnosis, while other cancers can be successfully treated [2]. One way of tracking cancer progression is to assess gene expression differences across the multi-stage process of cancer development. To our knowledge, limited research has focused on the progression of cancer in relation to gene expression. The numerous genetic changes that accrue over the course of early-stage to advanced-stage cancer development can be traced by RNA-Seq.

RNA-Seq is a high-throughput sequencing technology with computational methods to determine the quantity of RNA present in a biological sample. The method examines the continuously changing cellular transcriptome, allowing for an efficient and comprehensive description of gene expression profiles between different conditions over time [3]. RNA-Seq data is often in the format of a gene-by-sample count matrix, with genes in rows, and samples along the columns. The elements in the matrix report for each sample, the number of reads that could be uniquely aligned to a particular gene. The raw read counts have to be adjusted or "transformed" to aid our understanding of cancer progression.

To demonstrate our approach to investigating RNA-Seq cancer progression over the course of early-stage to advanced-stage cancer, we illustrate a bar graph of a single raw count gene expression profile in two cancer types (Fig 1). The dark blue and light blue bars represent advanced-stage and early-stage cancer gene expression, respectively, for gene $x$. In advanced-stage, gene $x$ shows an identical expression profile in cancer types 1 and 2. Based on the same raw expression value, both cancer types will group together. However, when considering the early-stage gene expression profiles in both cancer types, it's worth noting that the difference in expression between advanced-stage and early-stage cancer gene expression in cancer type 1 is greater than the difference in cancer type 2.

The present study aims to normalize advanced-stage with early-stage RNA-Seq data to investigate cancer progression in relation to gene expression. The normalization method corrects for genes that display less expression variability in advanced-stage cancer samples but display a high variability in early-stage cancer samples. As a result, more meaningful information is available in which the two distinct cancer types can be differentiated based on the differences in gene expression profiles, or cancer progression, from early-stage to advanced-stage cancer. The development of such high-throughput genome analysis techniques for research on cancer has a significant impact on clinical treatment, as the discovery of cancers that differentiate in

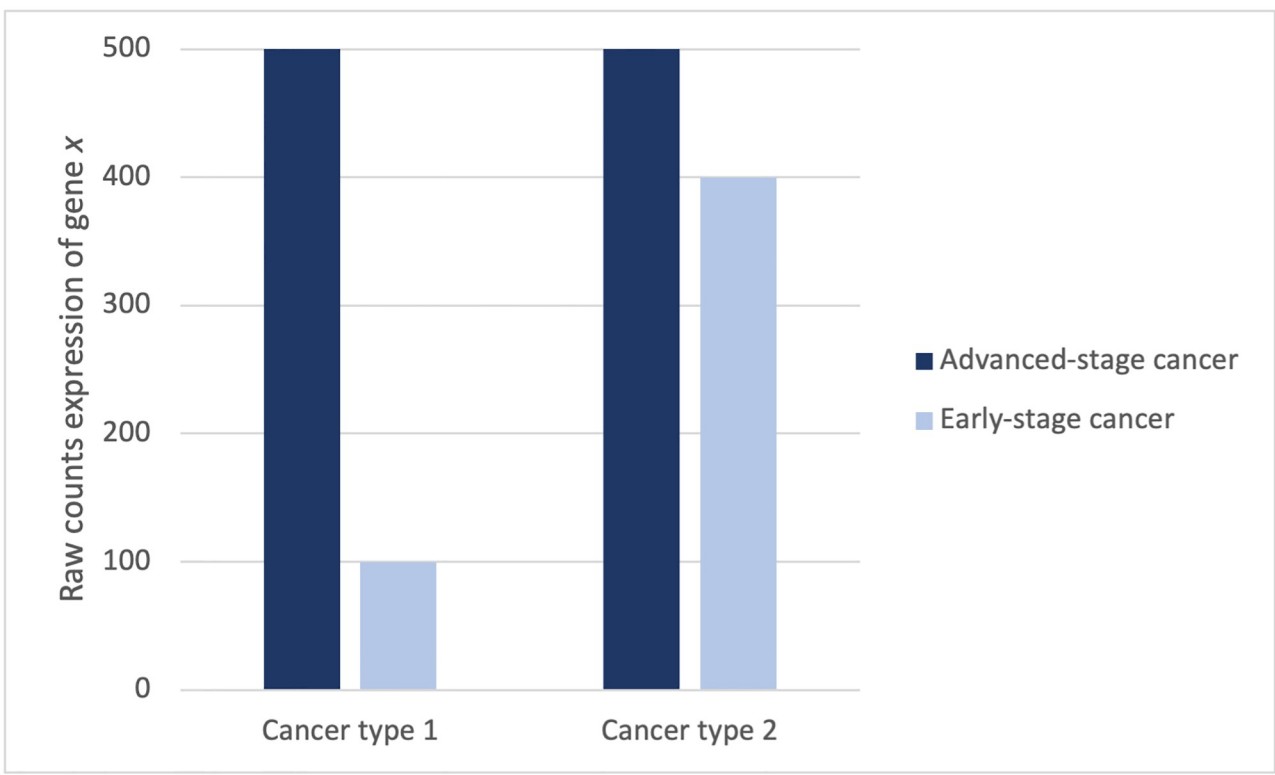

**Fig 1. Raw RNA-Seq data of advanced-stage and early-stage gene expression of gene *x* in two cancer types.** Cancer type 1 and cancer type 2 show a gene expression fold increase of 4 and 1, respectively, from early to advanced-stage cancer.

gene expression profiles (subtypes) is useful for guiding clinical treatment of multiple cancer [4].

The normalization method evaluated was performed by Frost and colleagues [5]. This method involves calculating the quotient of cancerous samples (dividend) and normal/non-cancerous samples (divisor), thereby producing normalized differential RNA expression profiles within a specific condition. However, many RNA-Seq research projects do not generate normal sequenced samples. Accordingly, we propose that early-stage cancer samples be used. We further hypothesize that using early-stage cancer samples will provide a more accurate genetic landscape of the multi-stage cancer progression.

## 2. Materials and methods

### 2.1. Data acquisition and processing

Cancer progression was investigated in early-stage and advanced-stage cancer. The datasets examined were selected based on cancers known to have an increased survival risk among patients due to associated autoimmune diseases. This includes five cancers; Diffuse Large B-cell Lymphoma (DLBCL) [6, 7], Lung Cancer [8], Cervical cancer [9], Liver cancer [10, 11], and Testicular cancer [12].

RNA-sequenced gene expression profiles for both early- and advanced-stage cancer were downloaded from the UCSC Xena database using cancer-specific data from The Cancer Genome Atlas cohort, from the Genomic Data Commons (GDC-TCGA) [13] (Table 1). Each patient's expression profile was organized in a gene-by-sample genomic matrix.

**Table 1. Cancer datasets.** The cancer cohorts were limited according to clinical or tumor stage and the primary site involved in each cancer. Patient samples were categorized in early-stage and advanced-stage, as well as the primary sites.

| | | Number of samples | |
|---|---|---|---|
| **Cancer cohort** | **Primary site** | **Early-stage** | **Advanced-stage** |
| Diffuse Large B-cell Lymphoma | Lymph Node | 4 | 8 |
| Lung Adenocarcinoma | Bronchus and Lung | 28 | 28 |
| Cervical Cancer | Cervix uteri | 8 | 22 |
| Liver Cancer | Liver and intrahepatic bile ducts | 20 | 6 |
| Testicular Cancer | Testis | 15 | 15 |

Additional metadata includes the associated phenotypic and survival profiles of each patient (S1 File).

The cancer datasets were made up of 60,483 unique Ensembl identifiers, which included transcript-non-specific expression data for all coding genes plus long non-coding RNA (lncRNA), pseudogenes, and multiple forms of non-coding transcripts [14]. The datasets quantified gene expression as $\log_2(x+1)$ with x referring to the count of reads mapped to a specific genetic region in the human reference genome (GRCh38.p2, gencode release 22).

Genes having ENSG identifiers annotated with a protein-coding biotype were extracted using Ensembl BioMart (GRCh38.p13, Ensembl 104, May 2021) [15]. This eliminated 40,927 (67,7%) non-coding entries leaving 19,556 protein-coding entries. The gene expression of the 19,556 protein-coding genes as $\log_2(x+1)$ was converted to raw counts for further analysis, as it was found that raw RNA-Seq data may perform better for capturing more original transcriptome patterns in different disease conditions [16].

## 2.2. Data normalization

The normalization method involves calculating the quotient of advanced-stage gene expression and early-stage gene expression (GitHub code: https://github.com/3270006/tracking-cancer-progression). We followed the same calculations established by [5].

**2.2.1. Gene and tissue correction.** The gene-by-sample matrices from each cancer cohort in Table 1 were used to assemble early-stage (E) and advanced-stage (A) gene expression matrices. This included:

A, s×q matrix for advanced-stage gene expression and,

E, s×r matrix for early-stage gene expression.

Where q and r represent the number of advanced-stage and early-stage cancer samples, respectively, and s the number of protein-coding genes.

Two binary primary site classification matrices were created for each gene expression matrix. This included:

P$^A$, t×q matrix for advanced-stage cancer primary sites and,

P$^E$, t×r matrix for early-stage cancer primary sites.

Where q and r represent the number of advanced-stage and early-stage cancer samples, respectively, and t the number of primary sites.

The advanced-stage cancer expression vector of gene *i* in matrix A was multiplied by the binary classification vector for primary site *I* in matrix P$^A$ as shown in Eq 1, resulting in a

vector of tissue-specific advanced-stage cancer gene expression $X_i$.

$$X_i = P^A_I \odot A_i \tag{1}$$

The early-stage expression vector of gene *i* in matrix E was multiplied by the binary classification vector for primary site *I* in matrix $P^E$ as shown in Eq 2, resulting in a vector of tissue-specific early-stage gene expression $Y_i$.

$$Y_i = P^E_I \odot E_i \tag{2}$$

$X_i$ and $Y_i$, were computed based on the series of vectors of all primary sites and all protein-coding genes to build three-dimensional matrices for X (advanced-stage cancer) and Y (early-stage cancer). The $X_{i,j,I}$ three-dimensional matrix represents the raw count gene expression value for gene *i* in advanced-stage cancer j of primary site *I*. While, the three-dimensional matrix of $Y_{i,k,I}$ represents the raw count gene expression value for gene *i* in early-stage cancer k of primary site *I*.

The initial phase of calculating for the normalized dataset (subsequently called 'Tissue-corrected'), involved creating a mean normalized expression $G^{tissue}$ for gene *i* at each primary site *I*, as given in Eq 3. To summarize, the sum of early-stage gene *i* within each primary site *I* was calculated.

$$G^{tissue}_{i,I} = \frac{1}{m_I} \sum_{k=1}^{r} Y_{i,k,I} \tag{3}$$

Where r is the number of early-stage cancer samples in primary site *I*. The calculation to determine for $m_I$ are shown in Eq 4, where the sum of a given primary site in the binary matrix $P^E$ were calculated for all early-stage samples.

$$m_I = \sum_{k=1}^{r} P^E_{k,I} \tag{4}$$

Finally, the tissue-corrected gene expression matrix $L^{tissue}$ was calculated as shown in Eq 5.

$$L^{tissue}_{i,j,I} = ln\left(\frac{X_{i,j,I}}{G^{tissue}_{i,I}}\right) \tag{5}$$

## 2.3. Weighted gene co-expression network analysis

Both the advanced-stage cancer gene expression as raw count (uncorrected) and the normalized tissue-corrected datasets were analysed. The 19,556 protein-coding genes were subjected to Weighted Gene Co-expression Network Analysis (v. 1.70–3) (WGCNA) R package [17, 18].

**2.3.1. Data pre-processing.**   The uncorrected matrix was filtered of genes that had a count of less than 10 in more than 90% of samples as recommended by the WGCNA authors, resulting in 17,436 protein-coding genes. The tissue-corrected matrix was filtered by removing all genes that had a row sum of zero, resulting in 19,350 protein-coding genes.

**2.3.2. Gene co-expression network construction.**   To construct a weighted network, a correlation matrix between each pair of genes across all samples was calculated. A soft threshold power β was calculated to amplify the correlation between genes. The optimal power value was selected based on a scale-free topology criterion ($R^2 > 0.8$). Based on this, an adjacency matrix was constructed, followed by the generation of a topological overlap matrix (TOM), and computation of the corresponding dissimilarity (1-TOM) values [19, 20].

To group the protein-coding genes, an average linkage hierarchical clustering based on the *hclust* function in conjunction with the dissimilarity TOM was used, resulting in a gene hierarchical clustering tree (tree graph). A novel dynamicTreeCut algorithm (v. 1.63–1) was employed to identify the clusters, in which branches of the dendrogram were sliced to determine the modules. Modules represent the partitioning of protein-coding genes into distinct groups based on expression values co-correlated and variable across the cancer cohorts. Modules were named using the default WGCNA settings, which assign a colour to each module.

## 2.4. Pathways and transcription factor enrichment analyses

A popular gene set enrichment analysis tool, WebGestalt (WEB-based GEne SeT AnaLysis Toolkit) was used to extract biological information from genes of interest [21]. The over-representation analysis (ORA) in the WebGestaltR package (v. 0.4.4) was used to characterize the genes of interest that were grouped inside each module found by WGCNA [22–24]. The ORA used all protein-coding genes as a reference set, the WikiPathways [25, 26] and Kyoto Encyclopedia of Genes and Genomes (KEGG) [27] databases for functional annotations, and the Benjamini-Hochberg method for multiple testing correction [28].

Transcription factor (TF) enrichment analysis was performed on the genes of interest that were grouped inside each module found by WGCNA using the ChEA3 database webserver application [29]. To estimate the TF-target enrichment, the ARCHS4 resource were selected as it uses a co-expression method to compile a list of genes that are controlled by each TF.

## 2.5. Clustering by transcript profiling

The clustering of cancer samples is the most basic and exploratory analysis to find groups of samples sharing similar gene expression patterns, which can lead to the discovery of new cancer subtypes. Therefore, gene expression profiles will be subjected to clustering analysis to investigate the grouping of cancer samples. Accordingly, the computation model was used to predict cancer clusters (subtypes) that progressed differently and/or similarly.

The cosine distance between the expression profiles of the genes included in the modules and Ward's method for agglomeration were used to create clusters of similar cancers established by hierarchical clustering [30, 31]. The number of clusters was identified using the *find_k* function, which estimates *k* using maximal average silhouette widths [32]. This function forms part of the dendextend (v. 1.15.2) R package. Finally, the dendrograms were split into *k* groups to assign samples to a cluster.

## 2.6. Survival analysis

The genes categorized in each module by WGCNA across the clusters were subjected to a Cox regression model based on the Lasso algorithm of the glmnet R package (v. 4.1–3) [33–35]. The model reduces the number of candidate genes and selects the most significant genes for a patient's survival, assigning a regression coefficient value to each gene. The product of the coefficient value and the corresponding gene's expression value resulted in a prognostic risk score for each patient. The patient scores were used to calculate a median risk score. A status value of 1 or 0 was assigned to each patient based on whether the patient's score was above or below the median risk score. Kaplan-Meier (K–M) estimates for overall survival (OS) were generated according to the patient status information. The K–M curves were created using the *ggsurvplot* function from the survminer R package (v. 0.4.9).

## 2.7. Statistics

The statistical analysis was performed using the car (v. 3.0–11), DescTools (v. 0.99.43), and agricolae (v. 1.3–5) R packages. The statistics were conducted to evaluate for different gene expression in each module and primary sites across the clusters.

The differences in the gene expression were first evaluated for normality and equal variance using Shapiro-Wilk test of normality [36] and Levene's test of homogeneity [37], respectively. If the Shapiro-Wilk null hypothesis was not rejected ($P \geq 0.05$; $H_0$: normal distribution) and Levene's test null hypothesis were not rejected ($P \geq 0.05$; $H_0$: equal variance across groups), an analysis of variance (ANOVA) [38] was employed. If the ANOVA null hypothesis of equal mean gene expression in each module and primary site was rejected by chance ($P \leq 0.05$), a Tukey's post-hoc test was used for pairwise comparisons [39].

In the event that Levene's test null hypothesis was rejected ($P \leq 0.05$; $H_1$: difference in variances between groups) and Shapiro-Wilk test resulted in either normal ($P \geq 0.05$) or not normal distribution ($P \leq 0.05$), then the Kruskal-Wallis test [40] was used to evaluate for differences in the gene expression in each module and primary site across clusters. If the Kruskal-Wallis was rejected, it can be concluded that equal median gene expression across groups was rejected, a post-hoc analysis was performed using Dunn's test [41].

## 3. Results and discussion

Both the uncorrected and tissue-corrected matrices were evaluated to determine if the normalization method represents differences in the true gene expression. The normalization method is considered effective if the normalized gene expression has an increased power in differentiating samples based on cancer type and clinical and phenotypic information.

### 3.1. Uncorrected RNA-Seq

The uncorrected protein-coding genes were inserted into WGCNA. The soft-thresholding power was defined as 20, with a scale-free topological index of above 0.8. This resulted in a gene tree and corresponding module colours. Similar modules were merged using the associated adjacency heatmap. The merged modules and the number of genes in each module was used for further analysis (S1 Fig).

A total of 3175 genes were categorized into 32 modules using WGCNA. Of those, only 10 modules were enriched for functional pathway annotations with WikiPathways: brown, cyan, grey60, magenta, purple, dark green, dark grey, light cyan, light steel blue 1, and tan. The first five modules were enriched for tissue-specific processes (ORA, $P \leq 0.047$). The latter five modules were enriched for cancer-relevant processes (ORA, $P \leq 0.045$).

It was found that the tan module had the highest total genes detected in biological pathways. It was also noteworthy that a repetition of the same pathway description appeared in several different modules. The same behaviour was noted with KEGG pathway analysis (S2 Fig). This indicates that the uncorrected dataset, which did not undergo normalization, did not efficiently depict gene expression differences.

The hierarchical clustering of cancer samples using the 3175 genes resulted in two cancer clusters (Fig 2). The primary site composition of each cluster was evaluated to determine if each primary site corresponded to the cluster assignment. Both clusters were primary sites heterogeneous. Cluster 1 was composed of samples of DLBCL (13.2%), lung (35.8%), liver (5.7%), cervical (22.6%), and testicular cancer (22.6%). While cluster 2 was composed of DLBCL (3.8%), lung (34.6%), liver (11.5%), cervical (38.5%), and testicular cancer (11.5%). The uncorrected dataset failed to correctly segregate the cancer samples in different clusters (Fig 2).

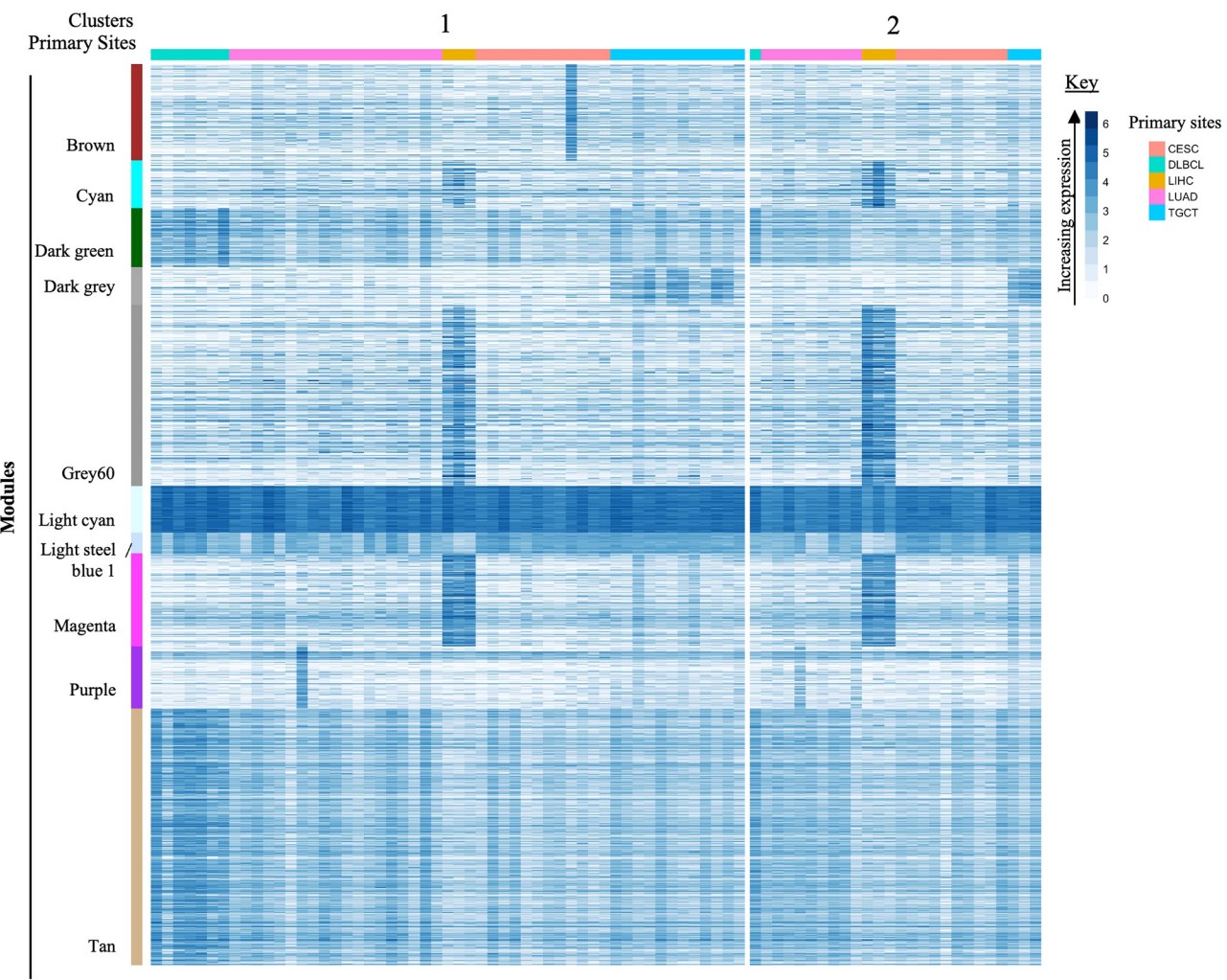

**Fig 2. Heatmap of uncorrected RNA-Seq data illustrating module expression within cancer clusters.** The colour bar on the left shows modules identified by WGCNA and enriched for functional pathway annotations. The rows are further composed of protein-coding genes with raw count values. Clusters of similar cancer cohorts are indicated across the top, and cancer cohorts are displayed by the colour bar along the top with the key on the right. *Primary sites abbreviations: CESC = Cervical squamous cell carcinoma; DLBCL = Diffuse Large B-cell Lymphoma; LIHC = Liver Hepatocellular Carcinoma; LUAD = Lung Adenocarcinoma; TGCT = Testicular Germ Cell Tumors.

The statistical analysis outlined in the methods section was performed to compare each module across the cancer clusters. From the 10 enriched modules, seven modules; cyan, dark green, dark grey, grey60, light cyan, light steel blue 1, and tan were characterized by significantly different expressions (Kruskal-Wallis $P \leq 0.0008$) across cancer clusters. While the magenta, purple (ANOVA, $P \geq 0.08$) and brown modules (Kruskal-Wallis, $P = 0.31$) did not show differential expression across clusters. That is, WGCNA selected genes with less differential power, because of non-normalization, resulting in heterogeneous clusters composed of samples from different primary sites (Fig 2).

The same statistical analysis was performed to compare each primary site in Cluster 1 to the equivalent primary site in Cluster 2 for each module. This computation was performed to determine if the segregation of primary sites into Clusters 1 and 2 was based on changes in the gene expression. The statistical test showed no significant difference between sample groups of the same primary sites from the two different clusters. It can be said that the clustering of the

uncorrected dataset failed to segregate the primary sites based on different gene expression. Evidently, the unnormalized genes failed to show differentiation.

## 3.2. Tissue-corrected RNA-Seq data

The tissue-corrected protein-coding genes were inserted into WGCNA. A soft threshold selection of the lowest β value that leads to $R^2 > 0.8$ was selected as 21. This resulted in a gene tree and corresponding module colours. Similar modules were merged using the associated adjacency heatmap. The merged modules and the number of genes in each module was used for further analysis (S3 Fig).

WGCNA identified 617 genes distributed into seven modules. The module that composed the most and least genes was the brown and pink modules, respectively. Of the seven modules, KEGG analysis enriched five modules (S4 Fig), while a total of four modules were found to be enriched for functional pathway annotations with WikiPathways. This included the black, brown, magenta and turquoise modules (Fig 3), of which all four modules were enriched for cancer-related processes (ORA, $P \leq 0.038$). The pathway descriptions identified in the four modules are indicated in the bar chart in Fig 3. Each colour bar represents the module colour and shows the number of genes that were enriched for that module. Analysing the degree of enrichment and terms further signifies the difference of each module.

The black module was enriched for cytoplasmic ribosomal proteins (ORA, $P < 0.001$). The brown module was enriched for NK cell, T cell or inflammatory signalling (ORA, $P \leq 0.021$). It was also found that the brown module has the highest total genes detected in biological pathways. The magenta module enriched for mRNA processing (ORA, $P < 0.0001$). Meanwhile,

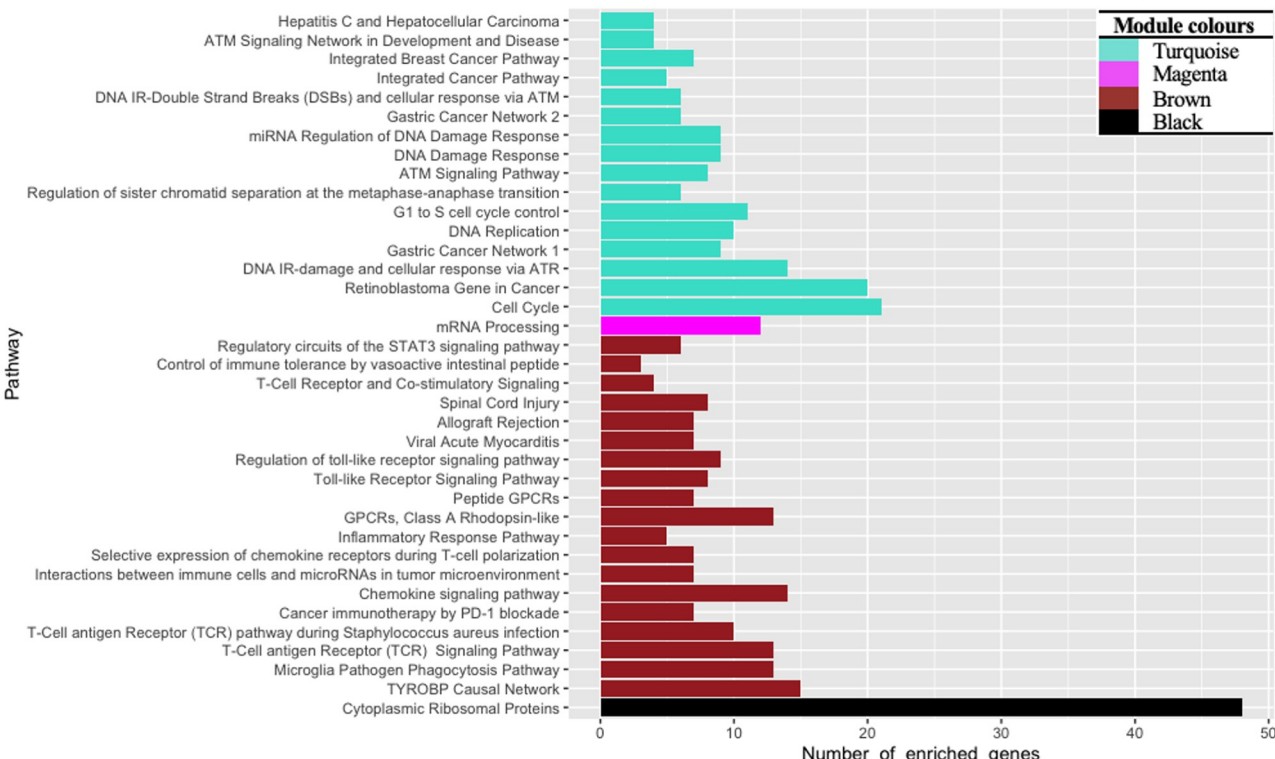

**Fig 3. WikiPathways enrichment of gene modules detected by WGCNA from the tissue-corrected dataset using the ORA, WebGestalt.**

processes relevant to the cell cycle progression were enriched in the turquoise module (Fig 3). The turquoise module was the largest module comprising 139 genes and also identified pathways that were related to other cancers such as breast cancer, gastric cancer, and retinoblastoma. Gastric adenocarcinoma has been reported to be correlated to the investigated cancers including liver carcinoma and lung cancer through specific genes [42]. It was noted that some genes were shared between the detected cancer pathways, this included the AURKA gene, which was involved in the gastric and breast cancer pathways. An increased gene expression of the AURKA gene has been previously identified in the liver and lung cancer [43]. Gastric and the retinoblastoma pathways further shared the MCM4, TOP2A and RFC4 genes, that have been reported in the studied cancers, where MCM4 is overexpressed in liver cancer [44], TOP2A promotes lung cancer [45], and RFC4 has a high expression in liver, lung, and cervical cancer [46].

Moreover, cancer progression and the retinoblastoma pathway are closely connected [47, 48]. It was found that the retinoblastoma and the breast cancer pathways shared the CHEK1 gene, a gene that has been reported in the development of human malignant tumors, such as lung and cervical cancers [49]. Therefore, the enriched module genes detected in the studied cancers could suggest that they play a role in cancer development and thus could also be relevant to other cancer types.

The WGCNA module genes were further subjected to TF enrichment analysis, to gain evidence for potential mechanistic connection of transcriptome changes to specific TFs. ChEA3 TF analysis revealed associations between the observed gene expression changes and involved TFs. The top 5 prioritized TFs for each module are presented in S2 File, with documented information about their biological involvement in the context of cancer (S2 File). The analysis confirms, with supported literature, several TF relationships with the multiple cancers evaluated in this study.

Hierarchical clustering of the 617 genes in WGCNA modules detected eight clusters characterized by distinct expression of the four enriched modules (Kruskal-Wallis Test, $P < 0.0001$) (Fig 4). Post hoc analysis by Dunn's Test to assess pairwise differences across clusters in each module showed differential expression for 21 of 28 cluster comparisons for the black module, 25 of 28 comparisons for the brown module, 24 of 28 comparisons for the magenta module, and 27 out of 28 comparisons for the turquoise module. The high proportion of pairwise cluster comparisons with significant differences highlights the distinctive expression patterns in each module across clusters.

The primary site composition of each cluster was evaluated to determine if the cancer primary site corresponded to the cluster assignment. Cluster 1 was primary site homogenous, composed of only DLBCL samples, while Cluster 2 was primary site heterogeneous, composed of DLBCL and liver samples. Clusters 3 and 4 were primary site homogenous, however shows a segregation of lung samples. The same was observed in Clusters 5 and 6 with cervical samples and Clusters 7 and 8 composed of testicular samples (Fig 4).

The associated metadata of the cancer samples were investigated to determine if distinct phenotypes could have caused similar cancer cohorts to partition into separate clusters in Fig 4. The DLBCL samples present in Cluster 1 show gene profiles that are more upregulated in comparison to the Cluster 2 DLBCL samples. In addition, it was noted that DLBCL samples in Cluster 1 showed a higher number of extranodal sites involvement ($\geq 2$), while those in Cluster 2 showed no or low number of extranodal sites involvement ($\leq 2$). Common sites of extranodal spread are lung, liver, kidney, and bone marrow [50]. It has also been reported that DLBCL can be involved in virtually any organ [51]. Therefore, the DLBCL Cluster 2 found grouped with liver samples is an interesting finding, given the high prevalence of secondary liver involvement by lymphoma including DLBCL and indicates advanced disease [52].

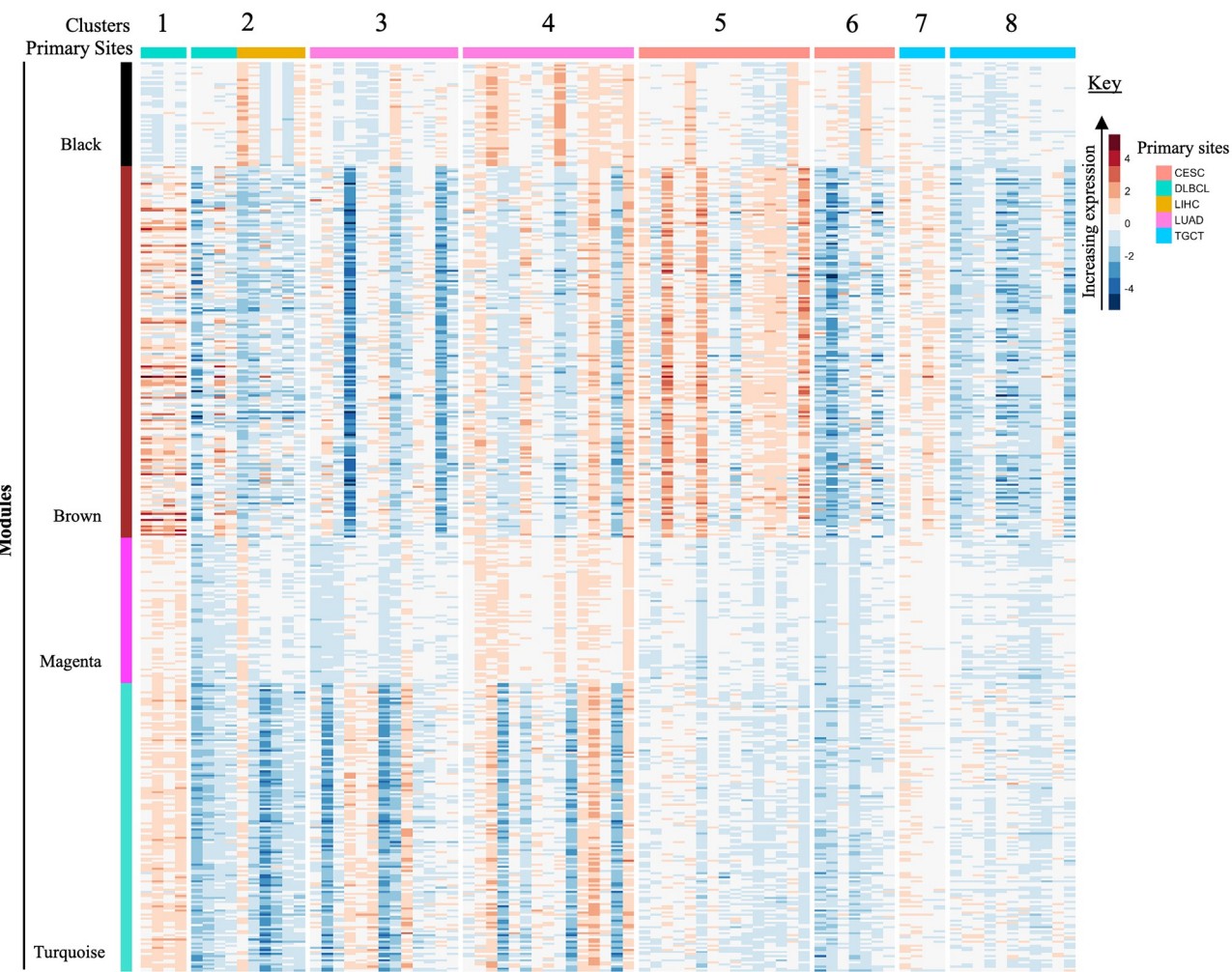

**Fig 4. Heatmap of tissue-corrected RNA-Seq data illustrating module expression within cancer clusters.** The colour bar on the left shows modules identified by WGCNA and enriched for functional pathway annotations. The rows are further composed of protein-coding genes with expression values obtained after data normalization. Clusters of similar cancer cohorts are indicated across the top and the cancer cohort are displayed by the colour bar along the top with the key on the right. *Primary sites abbreviations: CESC = Cervical squamous cell carcinoma; DLBCL = Diffuse Large B-cell Lymphoma; LIHC = Liver Hepatocellular Carcinoma; LUAD = Lung Adenocarcinoma; TGCT = Testicular Germ Cell Tumors.

However, this information of secondary liver involvement in the metadata associated to DLBCL is unavailable, and requires further investigation to support the claim that DLBCL patients have liver infection, as well as the use of a higher sample number, which was not possible for this study since the public data was not available. The phenotypic data for lung samples in Clusters 4 and 5 did not provide a clear reason for the segregation of the cancer cohort as some clinical information on the samples were incomplete.

It was discovered that the average overall survival of patients with cervical cancer represented in Cluster 5 were greater than the average overall survival of cervical cancer patients in Cluster 6. This led to a survival analysis in which the Cox regression model selected MAP4K1 (ENSG00000104814) categorized in the brown module as a prognostic gene. The upregulation of the MAP4K1 gene has been found to be favourable in cervical cancer [53, 54]. According to Kaplan-Meier results in a recent study, the high expression of the MAP4K1 gene was beneficial to cervical cancer patients [55]. Their research focussed on PDCD1, a gene that is most

typically related to its expression on tumor-infiltrating lymphocytes. Moreover, they showed that PDCD1 significantly co-expressed with the following 15 genes, whose high expression is beneficial for cervical cancer patients; MAP4K1, ACAP1, CST7, CXCR6, GPR171, GZMH, GZMK, P2RY10, RASAL3, SH2D1A, TBC1D10C, ZNF831, GZMM, JAKMIP1, and PSTPIP1 [55]. We compared their finding to the results of our normalization method and discovered the PDCD1 gene as well as the first 12 of the 15 genes were co-expressed within the brown module. This finding validates the normalization method in this study, as upregulation is observed in the brown module for Cluster 5, whereas the brown module in Cluster 6 mainly illustrates downregulation (Fig 4). The normalized gene expression of MAP4K1 in cervical patient samples from Clusters 5 and 6 were extracted from the brown module and shown in Fig 5.

We corroborate the previous findings [53–55] in that the upregulation of gene MAP4K1, in Cluster 5, is favourable in cervical cancer patients as shown by the Kaplan-Meier curve, in Fig 6. Cluster 5 presents a longer life expectancy than the patient samples in Cluster 6.

The brown module were further subjected to TF enrichment analysis using an established computational tool to offer a better understanding of the associations between the observed gene expression changes and TFs in the context of cervical cancer. The TFs that were associated with the MAP4K1 gene in which the TF was found to effect cervical cancer survival was extracted and documented (S3 File). Several co-expressed genes that also play a role in cervical cancer survival identified in [55] were also linked to the TFs and highlighted (S3 File).

Lastly, the phenotypic data of testicular cancer, divided in Clusters 7 and 8, showed that the primary diagnosis of the patients in Cluster 7 was seminomas, while Cluster 8 were made up of patient samples that were primary diagnosis with type embryonal carcinoma testicular cancer, mixed germ cell tumor or Teratoma malignant.

To further demonstrate the significance of late-stage cancer samples normalized with early-stage cancer samples, an investigation was carried out with a normal tissue expression dataset from the Genotype-Tissue Expression (GTEx) Portal [56]. Normalized gene expression profiles using normal tissue samples were clustered and allowed for the segregation between distinct cancer types (S5 Fig). However, it failed to provide in-depth clustering based on subtypes within cancer types. As a result, the variations in gene expression, such as in cervical cancer that was associated with survival, could not be stratified by normalizing late-stage cancer samples with normal tissue. The results obtained with our method by normalizing late-stage with early-stage cancer samples demonstrate the ability of the method to cluster samples by cancer progression, rather than simply by cancer type as with the use of normal samples.

## 4. Conclusion

The RNA-Seq read count before normalization showed discrepancies in comparison to normalized gene expression. The goal of our normalization method was achieved, since it shows that advanced-stage cancer gene expression data can be normalized using early-stage cancer gene expression data. WGCNA analysis validated the results of the tissue-corrected matrix as the correct relationships between normalized gene expression were presented. It was further illustrated that the biological information was preserved and allowed more meaningful comparisons of each cancer cohort, including survival analyses.

The benefit of the normalization method used in the present study was twofold; (i) it was able to segregate tumor samples with different and similar progression, (ii) and it could cluster samples from distinct cancer types as well as samples within the same cancer type. A significant result of the latter was in the case of cervical cancer, in which gene MAP4K1 was segregated based on the gene's prognosis. This discovery demonstrated that the normalization method

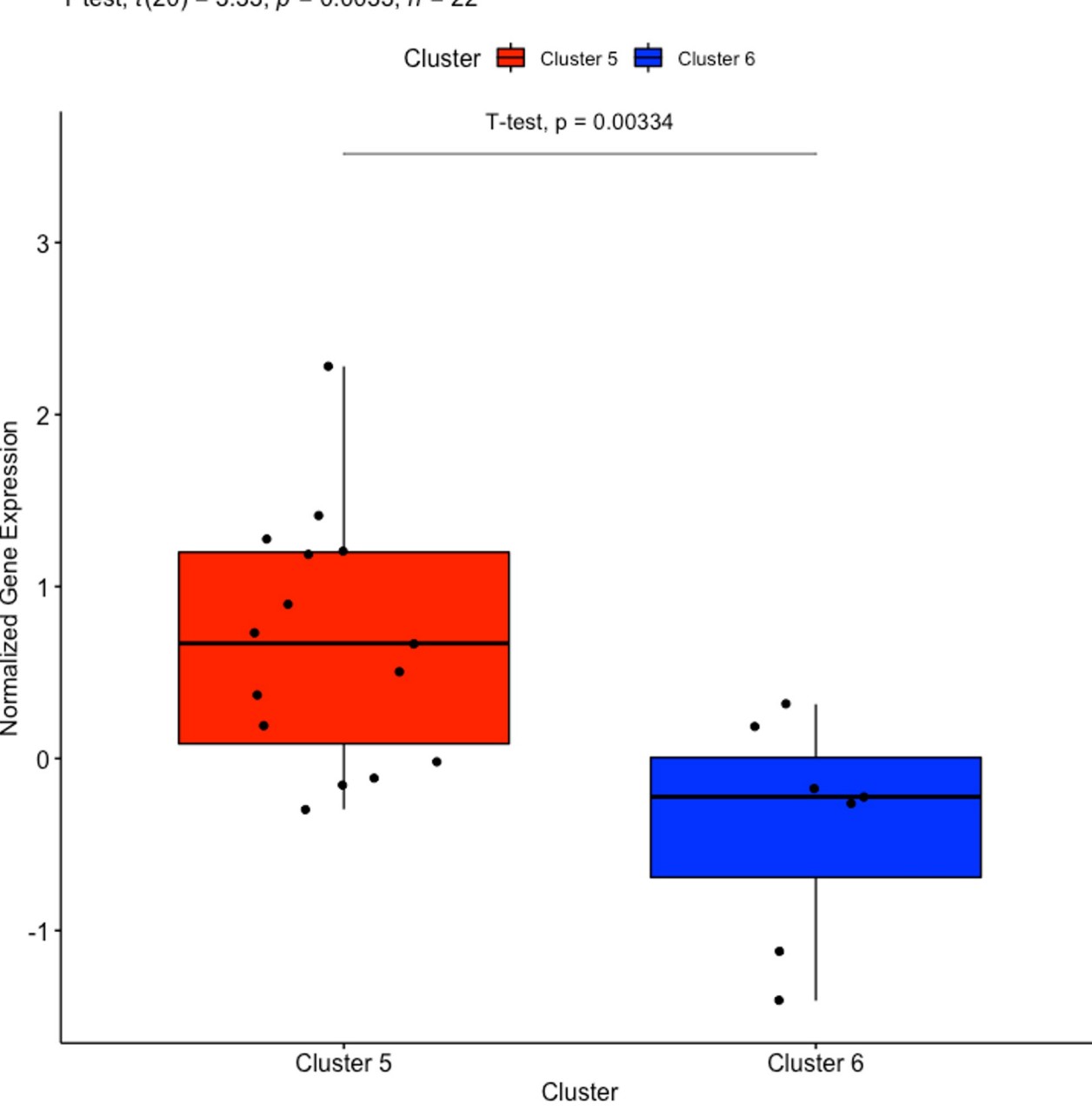

**Fig 5. Boxplot of gene MAP4K1 from cervical cancer samples categorized in the brown module by WGCNA.** The red box plot, constructed with Cluster 5 samples, shows upregulation of gene MAP4K1, while the blue box plot, constructed with Cluster 6 samples, shows downregulation of MAP4K1 gene.

can be used in conjunction with cancer clustering to identify areas of higher cancer risk as well as the cause of the increased risk.

The value of this method thus aids with hypotheses that seek to explore various novel cancer subtypes that segregate by different gene expression profiles and further investigate the biological association, clinical, or prognostic features linked to the cancer subtypes (clusters). Additionally, hypotheses that investigate cancer progression and identify cancer subtypes with

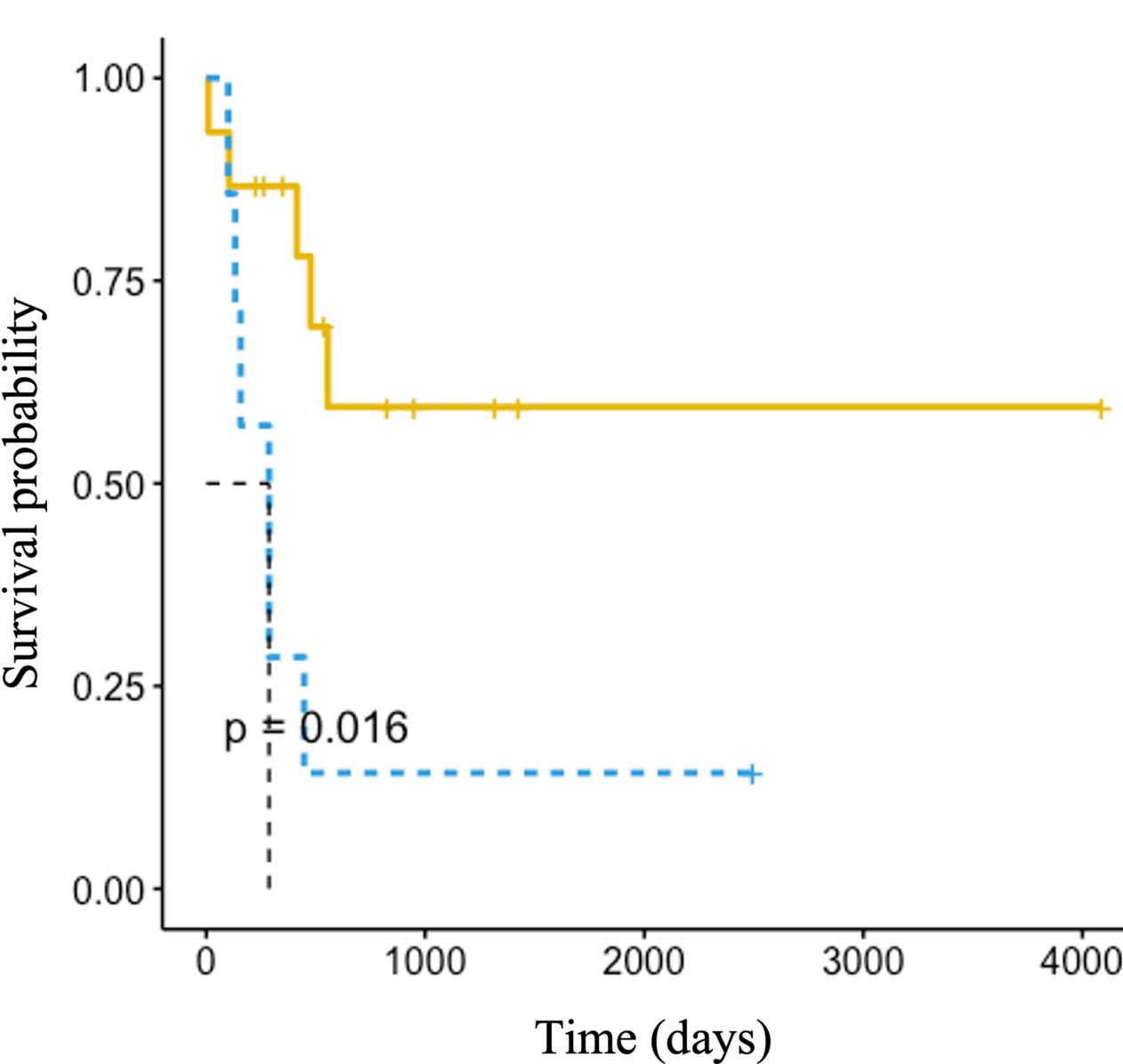

**Fig 6. Kaplan-Meier of MAP4K1 gene in cervical cancer patients.** Analysis shows the correlation between normalized gene expression level and patient survival in days. Patients were divided as detected in Clusters 5 and 6 after clustering according to transcript profiling.

different progression. New users can further use this method to find new subtypes in their data and associate it with the clinical data that they have.

## Supporting information

**S1 Fig. Uncorrected RNA-Seq data were inserted into WGCNA to identify gene modules.** (A) Soft threshold power. (B) Gene clustering tree. Each colour underneath the dendrogram shows the module assignment, and branches above represent the genes. The dynamic tree cut

shows the initial module detection and merged dynamic indicates the modules divided according to their similarity. (C1) Module eigengene dendrogram identified groups of correlated modules. The red line indicates the module eigengene threshold of 0.25 and (C2) Eigengene adjacency heatmap of different gene co-expression modules. In the heatmap, the blue colour represents low adjacency, while the red represents high adjacency. (D) Barplot of 32 co-expression modules constructed after similar modules were merged with module size at the top of each bar.
(TIF)

**S2 Fig. KEGG enrichment of gene modules detected by WGCNA from the uncorrected RNA dataset using the ORA, WebGestalt.**
(TIF)

**S3 Fig. Tissue-corrected dataset were inserted into WGCNA to identify gene modules.** (A) Soft threshold power. (B) Gene clustering tree. Each colour underneath the dendrogram shows the module assignment, and branches above represent the genes. The dynamic tree cut shows the initial module detection and merged dynamic indicates the modules divided according to their similarity. (C1) Module eigengene dendrogram identified groups of correlated modules. The red line indicates the module eigengene threshold of 0.25 and (C2) Eigengene adjacency heatmap of different gene co-expression modules. In the heatmap, the blue colour represents low adjacency, while the red represents high adjacency. (D) Barplot of seven co-expression modules constructed after merged modules with module size at the top of each bar.
(TIF)

**S4 Fig. KEGG enrichment of gene modules detected by WGCNA from the tissue-corrected RNA dataset using the ORA, WebGestalt.**
(TIF)

**S5 Fig. Heatmap of tissue-corrected RNA-Seq data of late-stage cancer samples normalized with normal tissue samples, illustrating module expression within cancer clusters.** Normal tissue expression dataset was obtained from the Genotype-Tissue Expression (GTEx) Portal. To match the number of male/female ratios as in the late-stage cancer samples, the same number normal tissue samples of male/female ratios were randomly selected, except for cervical cancer, which only had 10 normal tissue samples. The colour bar on the left shows modules identified by WGCNA and enriched for functional pathway annotations. The rows are further composed of protein-coding genes with expression values obtained after data normalization. Clusters of similar cancer cohorts are indicated across the top and the cancer cohort are displayed by the colour bar along the top with the key on the right. *Primary sites abbreviations: CESC = Cervical squamous cell carcinoma; DLBCL = Diffuse Large B-cell Lymphoma; LIHC = Liver Hepatocellular Carcinoma; LUAD = Lung Adenocarcinoma; TGCT = Testicular Germ Cell Tumors.
(TIF)

**S1 File. The data underlying the results presented in this study are publicly accessible from the UCSC Xena data browser (https://xenabrowser.net) from individual cancer cohorts.**
(PDF)

**S2 File. Top 5 TFs derived from the ChEA3 enrichment analysis of each tissue-corrected WGCNA module.** The biological role indicates the role of the identified TF in cancer according to literature.
(PDF)

**S3 File. Transcription factors (TFs) enrichment analysis of tissue-corrected WGCNA brown module.** A list of TFs and their corresponding rank according to ARCHS4 co-expression, with documented information about their biological function associated with survival in the context of cervical cancer. The genes in bold were previously found [55] to play a role in cervical cancer survival.
(PDF)

## Author Contributions

**Conceptualization:** Michelle Livesey, Sophia Catherine Rossouw, Hocine Bendou.

**Formal analysis:** Michelle Livesey.

**Funding acquisition:** Alan Christoffels, Hocine Bendou.

**Investigation:** Hocine Bendou.

**Methodology:** Michelle Livesey, Renette Blignaut, Hocine Bendou.

**Project administration:** Hocine Bendou.

**Supervision:** Hocine Bendou.

**Writing – original draft:** Michelle Livesey.

**Writing – review & editing:** Sophia Catherine Rossouw, Renette Blignaut, Alan Christoffels, Hocine Bendou.

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
