## [Decision Letter · Decision Letter 0]

7 Oct 2022

PONE-D-22-16548Tracking cancer progression: A normalization method to identify differential RNA-Seq gene expression in the multi-stage early to advanced-stage cancer development.PLOS ONE

Dear Dr. Bendou,

Thank you for submitting your manuscript to PLOS ONE. After careful consideration, we feel that it has merit but does not fully meet PLOS ONE’s publication criteria as it currently stands. Therefore, we invite you to submit a revised version of the manuscript that addresses the points raised during the review process.

Please consider carefully the comments of the reviewers. Particularly relevant is to highlight the advantages of your proposed method and its contrast with the one presented in reference 4, as well as the biological relevance of your findings so that the full extent of your contribution will be clearer.

We look forward to receiving your revised manuscript.

Kind regards,

Enrique Hernandez-Lemus, Ph.D.

Academic Editor

PLOS ONE

Journal Requirements:

Reviewers' comments:

Reviewer's Responses to Questions

**Comments to the Author**

1. Is the manuscript technically sound, and do the data support the conclusions?

Reviewer #1: Yes

Reviewer #2: Partly

2. Has the statistical analysis been performed appropriately and rigorously? 

Reviewer #1: I Don't Know

Reviewer #2: Yes

3. Have the authors made all data underlying the findings in their manuscript fully available?

Reviewer #1: Yes

Reviewer #2: Yes

4. Is the manuscript presented in an intelligible fashion and written in standard English?

Reviewer #1: Yes

Reviewer #2: Yes

5. Review Comments to the Author

Reviewer #1: The authors show that by normalizing the gene expression from late stage cancer using early stage cancer samples to track progression.

The normalization method has been applied before using non tumor samples to normalize instead of early stage samples as in this case.

Minor suggestions/comments:

1. Figure 1 could include depiction of foldchange sizes in the two cancer types.

2. Most, if not all, of the subfigures in Figure 2 should be supplementary figures, sames as with Figure 5.

3. In the methods section, in the vector multiplication it needs to be clarified which operation is being applied; the notation implies cross product, but probably the one being used is element-wise multiplication (Hadamard product).

4. Including the code for normalization and analysis steps would improve the paper, it would clarify steps, and would be of use to the commuity. The code could be included as either supplementary or with a github (or similar link.

5. An explanation of the rational for the differential expression steps would be very useful. Why not use a standard differential expression analysis package such as DeSEQ2 or EdgeR?

6. It would be interesting to know if using early stage works like a surrogate of normal tissue (as in the original publication for the normalization method) or it is actually better or gives a different insight.

Reviewer #2: The manuscript by Livesey et al describes a computational method aimed at the identification of gene expression signatures associated with cancer development from early to advanced stages. A central approach is to analyze multiple cancer types from multiple tissue sites, normalize all expression values for each gene at advanced cancer stage by the average expression value at the early stage, and apply the WGCNA method to identify gene modules corresponding to cancer progression. This method is a modification of a previously described approach (reference 4) to the identification of gene expression signatures in various combined cancer types compared to normal tissue samples. The method proposed in this manuscript is methodologically identical to the “tissue-corrected” normalization approach of reference 4, only applied to the comparison of samples from early vs late stages of cancer. The authors describe the application of their method to publicly available gene expression data from early and late stages of five cancer types, with 4 to 28 samples in each group. They compared the WGCNA results based on original expression data to the results based on the normalized data and observed a stronger pathway enrichment in WGCNA gene modules and a more clinically relevant clustering by sample type based on the normalized data. The manuscript is clearly written, with the exception of a few minor passages.

My general concerns are the novelty of the proposed methodology, the evaluation approach, and a very minimal discussion of the actual known and new gene signatures detected by this method.

Major comments:

1. The direct advantage of this method to the discovery of known and new markers of advanced cancer stages is not described to a full extent. Detecting MAP4K1 (Fig. 8,9) whose upregulation is known to be associated with better outcomes is an interesting finding. Could this finding be made by a more basic differential expression analysis between early and late stages of cervical cancer? Does this method detect other markers of late cervical cancer?

More generally, it is not fully clear how the analysis of multiple cancers and the use of WGCNA modules bring new specific predictive gene markers.

2. It would be important to put these findings in the context of

a. known progression of gene expression in these specific cancers described in the literature, and

b. more basic approach of simply detecting differentially expressed genes (DEGs) between early and late stage for each cancer type in the evaluation dataset.

3. Lines 420-421:

“The clustering of DLBCL and liver samples in Cluster 2, could suggest that the two cancers progress in a very similar manner.”

This grouping of two cancers in the same cluster is not a sufficiently strong argument to make this far-reaching conclusion. To make a more compelling argument, it would be important to analyze and interpret specific gene signatures that are shared between progression of these two very different cancers from very different tissue environments.

4. Conclusions about a clinically relevant clustering of samples from different subtypes of testicular cancer (lines 448-451):

Could these and similar conclusions about grouping of cancer subtypes be achieved by a more basic DEG analysis and clustering these samples within each cancer type by the expression of DEGs?

5. The rationale behind the choice of a relatively limited evaluation dataset was not fully described. Lines 108-110:

“The datasets examined were selected based on cancers known to have an increased survival risk among patients due to associated autoimmune diseases.”

The numbers of these patient samples (Table 1) are relatively small.

Why this selection of cancer types was made? Would autoimmune signatures contribute as an unrelated confounding factor to late-stage cancer expression signatures?

6. All conclusions about functional gene enrichment in detected WGCNA modules are based on the WikiPathway database. WikiPathway is only one of the public pathway databases, and arguably not the best one. It would be beneficial to analyze the enrichment of pathways from other more established resources, e.g. MSigDB, KEGG, Reactome, ChEA gene categories, as well as TF LOF and TF perturbation gene collections available from EnrichR.

7. Although some of the pathways in Fig. 6 are general pan-cancer gene sets, e.g. pathways related to cell cycle or DNA damage, some of these pathways correspond to specific cancers (gastric cancers, breast cancer, etc) that are unrelated to the analyzed cancer types. This inconsistency and specific genes detected in these unrelated cancer pathways need to be addressed.

Minor comments:

1. Lines 126-127: Xena database provides not read counts but TPM values. x should be noted as TPM.

2. Text labels in Figures 2,3,5 are not legible.

3. Language needs to be clarified in a few passage throughout the text, e.g. lines 425-426:

“the average overall survival rate of patients with cervical cancer represented in Cluster 5 lived longer than the patients with cervical cancer in Cluster 6”

4. Heatmaps and color bars in Fig. 3,4,7: Coloring modules by bars and labeling these modules in text by the same color name is redundant. It would be more informative to have at least a very general and brief text labels by the functions of these modules.

6. PLOS authors have the option to publish the peer review history of their article (what does this mean?). If published, this will include your full peer review and any attached files.

Reviewer #1: No

Reviewer #2: No

---

## [Author Response · Author response to Decision Letter 0]

7 Nov 2022

We have uploaded the Response to Reviewers document in the Attach Files section.

---

## [Decision Letter · Decision Letter 1]

8 Dec 2022

PONE-D-22-16548R1A normalization method to track cancer progression with RNA-Seq gene expression in the multi-stage early to advanced-stage cancer developmentPLOS ONE

Dear Dr. Bendou,

Thank you for submitting your manuscript to PLOS ONE. After careful consideration, we feel that it has merit but does not fully meet PLOS ONE’s publication criteria as it currently stands. Therefore, we invite you to submit a revised version of the manuscript that addresses the points raised during the review process.

 Please present a point-by-point response to all the comments by Reviewer 2 and modify your manuscript accordingly.

We look forward to receiving your revised manuscript.

Kind regards,

Enrique Hernandez-Lemus, Ph.D.

Academic Editor

PLOS ONE

Reviewers' comments:

Reviewer's Responses to Questions

**Comments to the Author**

1. If the authors have adequately addressed your comments raised in a previous round of review and you feel that this manuscript is now acceptable for publication, you may indicate that here to bypass the “Comments to the Author” section, enter your conflict of interest statement in the “Confidential to Editor” section, and submit your "Accept" recommendation.

Reviewer #1: All comments have been addressed

Reviewer #2: (No Response)

2. Is the manuscript technically sound, and do the data support the conclusions?

Reviewer #1: Yes

Reviewer #2: Yes

3. Has the statistical analysis been performed appropriately and rigorously? 

Reviewer #1: Yes

Reviewer #2: Yes

4. Have the authors made all data underlying the findings in their manuscript fully available?

Reviewer #1: Yes

Reviewer #2: Yes

5. Is the manuscript presented in an intelligible fashion and written in standard English?

Reviewer #1: Yes

Reviewer #2: Yes

6. Review Comments to the Author

Reviewer #1: The authors have addressed satisfactorily my comments and I appreciate that the authors made their code available.

Reviewer #2: The paper describes application of a previously published method proposed (“tissue-corrected” normalization approach of reference 4) to the comparison of samples from early vs late stages of cancer. Applying an existing method in a somewhat different context does not amount to a new methodological development.

My few main concerns have not been fully addressed in the revised manuscript.

1. This manuscript does not include any benchmarking, apart from showing an interesting but anecdotal consistency with previously reported biological results, even in the revised version, which describes a biologically meaningful clustering of 12 PDCD1 associated genes. A more rigorous large-scale methodological evaluation and comparison with other computational approaches would be important.

2. To show the value of the method to potential users, in addition to valuable examples of confirming previously known gene associations, it would be important to illustrate how this method can help generate new hypotheses of previously uncharacterized biological associations or mechanisms.

Based on the authors' response about the rationale for using a small dataset focused on a particular cancer type, this manuscript almost seems a pilot to a bigger project focused on HIV effects on Diffuse Large-B cell Lymphoma. This bigger project, hopefully with a larger patient cohort, larger body of results and hopefully new mechanistic hypotheses, may warrant a publication in this journal.

Minor comments:

1. My question about the interaction of cancer-specific signatures and autoimmune responses was not addressed by the authors:

Would autoimmune signatures contribute as an unrelated confounding factor to late-stage cancer expression signatures?

2. “Therefore, the use of the WikiPathway database was sufficient in providing the knowledge we required in the context of this specific investigation."

Using multiple databases of functional gene categories in addition to WikiPathways would be important, especially in the context of this manuscript where functional enrichment is a major source of confirming that results are reasonable. Some of these additional databases include more mechanistic knowledge, e.g. empirically validated TF targets, which would be beneficial for generating new mechanistic hypotheses based on the method's results.

7. PLOS authors have the option to publish the peer review history of their article (what does this mean?). If published, this will include your full peer review and any attached files.

Reviewer #1: No

Reviewer #2: No

---

## [Author Response · Author response to Decision Letter 1]

30 Jan 2023

Please see attached Response to Reviewers document.

---

## [Decision Letter · Decision Letter 2]

7 Mar 2023

PONE-D-22-16548R2A normalization method to track cancer progression with RNA-Seq gene expression in the multi-stage early to advanced-stage cancer developmentPLOS ONE

Dear Dr. Bendou,

Thank you for submitting your manuscript to PLOS ONE. After careful consideration, we feel that it has merit but does not fully meet PLOS ONE’s publication criteria as it currently stands. Therefore, we invite you to submit a revised version of the manuscript that addresses the points raised during the review process.

Some minor issues mostly related to the presentation are made by the reviewers (figures, p-value formats, etc.) please address them in your further revised version.

We look forward to receiving your revised manuscript.

Kind regards,

Enrique Hernandez-Lemus, Ph.D.

Academic Editor

PLOS ONE

Journal Requirements:

Reviewers' comments:

Reviewer's Responses to Questions

**Comments to the Author**

1. If the authors have adequately addressed your comments raised in a previous round of review and you feel that this manuscript is now acceptable for publication, you may indicate that here to bypass the “Comments to the Author” section, enter your conflict of interest statement in the “Confidential to Editor” section, and submit your "Accept" recommendation.

Reviewer #1: All comments have been addressed

Reviewer #2: All comments have been addressed

2. Is the manuscript technically sound, and do the data support the conclusions?

Reviewer #1: Yes

Reviewer #2: Yes

3. Has the statistical analysis been performed appropriately and rigorously? 

Reviewer #1: Yes

Reviewer #2: Yes

4. Have the authors made all data underlying the findings in their manuscript fully available?

Reviewer #1: Yes

Reviewer #2: Yes

5. Is the manuscript presented in an intelligible fashion and written in standard English?

Reviewer #1: No

Reviewer #2: Yes

6. Review Comments to the Author

Reviewer #1: Overall I think the manuscript lacks some clarity. I have the following comments:

- Figure 2 is not needed in my opinion. The point that it makes is that the same pathways were enriched in several modules, this could just be mentioned in the text.

- In line 75-82 the use of the word normalized is confusing. The authors give an introduction to RNA-seq and where they mention that raw counts need to be normalized to be used for further analysis. I do agree but normalizing raw counts reads is the normal standard practice, although in a different way in which authors propose normalization in this work. The paper in which they base their normalization method actually uses TPM which is a normalization strategy for RNA-seq. I think the authors need to address why they used unnormalized values (raw reads) as input instead of TPMs, and also point out in the introduction that they are proposing a normalization strategy that does not replace standard normalization but it is specific to their research question. Furthermore, comparing unnormalized gene expression of different samples makes no sense. For figure 1 they would have to explain that even normalized expression values such as tpm would still not address the issue depicted in Figure 1. The authors here propose normalization of gene expression for late stage cancer samples using early stage cancer samples. I understand what the author means but then need to clarify that their normalization method is not the same kind of normalization done to RNA-seq raw counts data such as accounting for library size.

- In line 244 the use of the word implemented is inaccurate. In computer science "implementing" implies writing code, not just executing or running a program with your own dataset.

- Most of the p values are not represented in a standard way (missing zeros before decimal points among others).

- I dont think screened is the appropriate verb here (line 360).

- Not sure what the next phase menas: "It was also found that the brown module has the highest enrichment". Enrichment score?, number of enriched pathways? pathways with the most genes?

- Modules can be enriched with certain pathways, but genes are not enriched, therefore all instances of "enriched genes" should be corrected.

- Line 401 "The WGCNA module genes were further subjected to TF" -> TF enrichment analysis

- "Moreover, they showed that PDCD1 significantly co-expressed with the following 15 genes, whose high expression is beneficial for cervical cancer patients; MAP4K1, ACAP1, CST7, CXCR6, GPR171, GZMH, GZMK, P2RY10, RASAL3, SH2D1A, TBC1D10C, ZNF831, GZMM, JAKMIP1, and PSTPIP1 [54]." Here the reference is a link to the human protein atlas for MAP4K1, so the reference only refers to the first gene and it seems like the authors claim it is for all 15 genes. If it is the case that they tried all 15 genes maybe they should include those figures as supplementary, for all 15 genes, or further explain in the text that they tested the genes in a website. As it currently stands readers expect to find a reference to a paper that claims that high expression of all 15 genes are beneficial for cervical cancer patients which is not what the reference is.

- In line 483 there is a claim that more than one tool was utilized for TF enrichment analyisis and I believe only one was used.

Reviewer #2: The authors addressed most of my comments in this new revision. As a response to the authors' question in the rebuttal, I suggest that functional pathway enrichment analyses based on KEGG database (Fig. 2 and 3 of rebuttal) should be included in the manuscript as supplementary figure(s). This is a minor recommendation that I trust the authors to follow through.

7. PLOS authors have the option to publish the peer review history of their article (what does this mean?). If published, this will include your full peer review and any attached files.

Reviewer #1: No

Reviewer #2: No

---

## [Author Response · Author response to Decision Letter 2]

22 Mar 2023

A Response to Reviewers document has been uploaded.

---

## [Editor Report · Decision Letter 3]

3 Apr 2023

Transforming RNA-Seq gene expression to track cancer progression in the multi-stage early to advanced-stage cancer development

PONE-D-22-16548R3

Dear Dr. Bendou,

We’re pleased to inform you that your manuscript has been judged scientifically suitable for publication and will be formally accepted for publication once it meets all outstanding technical requirements.

Kind regards,

Enrique Hernandez-Lemus, Ph.D.

Academic Editor

PLOS ONE
---

## [Editor Report · Acceptance letter]

13 Apr 2023

PONE-D-22-16548R3 

Transforming RNA-Seq gene expression to track cancer progression in the multi-stage early to advanced-stage cancer development. 

Dear Dr. Bendou:

I'm pleased to inform you that your manuscript has been deemed suitable for publication in PLOS ONE. Congratulations! Your manuscript is now with our production department. 

Kind regards, 

on behalf of

Prof. Enrique Hernandez-Lemus 

Academic Editor

PLOS ONE